

# Modelling detrital cosmogenic nuclide concentrations during landscape evolution in CIDRE V2.0

Sébastien Carretier, Vincent Regard, Youssouf Abdelhafiz, Bastien Plazolles[1]

[1]Geosciences Environnement Toulouse, GET, CNRS, IRD, CNES, Universite de Toulouse, 14 avenue E. Belin, F-31400, Toulouse, France

**Correspondence:** Sebastien Carretier (sebastien.carretier@get.omp.eu)

**Abstract.** The measurement of cosmogenic nuclide (CN) concentrations in riverine sediment has provided breakthroughs in our understanding of landscape evolution. Yet, linking this detrital CN signal and the relief evolution is based on hypotheses that are not easy to verify in the field. A model would help to better understand the statistics of CN concentrations in sediment grains. In this work, we present a coupling between the landscape evolution model Cidre and a model of the CN concentration in distinct

grains. These grains are exhumed and detached from the bedrock and then transported in the sediment to the catchment outlet with temporary burials and travel according to the erosion-deposition rates calculated spatially in Cidre. The concentration in the various CN can be tracked in these grains. Because the CN concentrations are calculated in a limited number of grains, they provide an approximation of the whole CN flux. Therefore, this approach is limited by the number of grains that can be handled in a reasonable computing time. Conversely, it becomes possible to record part of the variability in the erosion-

deposition processes in the grain-by-grain distribution of the CN concentrations by tracking the CN concentrations in distinct grains using a Lagrangian approach. We illustrate the robustness and limits of this approach by deriving the catchment-mean erosion from the [10]Be mean concentration of grains leaving a synthetic catchment uplifting at different rates and by comparing this derived erosion rate to the actual one calculated by Cidre.

## 1    Introduction

The concentration of cosmogenic nuclides (CN) produced in situ varies according to the depth of the minerals in which they are produced, their altitude, the stable or radioactive nature of the nuclide in question and the magnetic field (Gosse and Phillips, 2001). When a mineral in a grain is exhumed and then transported in rivers, the concentration of CN evolves by integrating stochastic variations of the residence times at different depths and altitudes over time. This integrative characteristic has been positively used to develop numerous approaches to quantify erosion-deposition processes averaged over catchment areas and

millennia (Schaefer et al., 2022). The most widespread approach is one that quantifies the average erosion rate of a catchment from its average CN concentration in a sand sample taken at its outlet (Brown et al., 1995). Since the pioneering work of Repka et al. (1997), other studies have explored the possibility of using the distribution of CN concentrations in distinct grains and possibly grains of different sizes to quantify erosion-deposition processes on hillslopes, in rivers and on alluvial deposits over periods of several thousand to millions of years (Braucher et al., 1998; Dunai et al., 2005; Gayer et al., 2008; Codilean et al.,



2008; Carretier et al., 2019). However, it is still difficult to link detrital CN concentration data to specific processes, whether on hillslopes or on a larger scale in river systems (Yanites et al., 2009). This difficulty arises from the stochasticity of grain transport, which can be temporarily stored and then recycled on different scales ranging from a flood event that erodes the banks of an alluvial river, to millions of years in the case of exhumation of ancient strata of a foreland basin.

To address this complexity, several models have been developed at the grain and catchment scales (Repka et al., 1997; Niemi et al., 2005; Codilean et al., 2008; Carretier et al., 2009, 2019; Ben-Israel et al., 2022), but without taking the evolution of the relief into account. At the same time, landscape evolution models (LEM) have made great progress by integrating an increasing number of processes, however their resolution over long time spans limits taking into account stochastic phenomena that require the intermittent storage of sediment grains on hillslopes (e.g. landslides) and along fluvial systems (e.g. sediment bars

and terraces). Carretier et al. (2016) proposed a coupling between the landscape evolution model in Cidre and the transport of tracer grains which are transported stochastically according to simple probability laws depending on the local erosion and deposition rates calculated on each cell of the LEM. This allowed the authors to highlight that some grains may have been stored for a long time before being recycled and evacuated by the rivers (Carretier et al., 2020). In this contribution, we present a development in Cidre to calculate the concentration of several CNs within the grains. To our knowledge, the only published

model that can jointly model the evolution of the relief and the evolution of the average CN concentration in the sediments is Badlands (Petit et al., 2023). In Badlands, this average concentration is calculated in proportion to the sediment fluxes from the different upstream sources on each grid cell. In Cidre, we adopt a different (Lagrangian) approach, which enables us to track the full distribution of CN concentrations in a population of grains.

We first present the Cidre model, including grain transport, and then describe the equations used to calculate the CN concentration over time in each grain. We show the accuracy and robustness of the algorithm by comparing the average catchment erosion rate calculated from the CN concentrations of outgoing grains with the true rate calculated in several simulations. In particular, we show that the average erosion rate is approximated to within 5% uncertainty in most of the cases with a limited number of grains.

## 2   The landscape evolution model Cidre

### 2.1   Mass balance equation

Cidre is a c++ code that solves the following mass balance on rectangular cells expressed here for the mean elevation $z$ of a cell through time $t$:

$$\frac{\partial z}{\partial t} \quad = \quad -\epsilon_r - \epsilon_h + D_r + D_h + U \tag{1}$$



This mass-balance refers to the Erosion("$\epsilon$")-Deposition("$D$") model (Davy and Lague, 2009) where the subscript 'r' ('river')
refers to erosion driven by flowing water and 'h' ('hillslope') to erosion due only to the topographic gradient or slope $S$. $U$ is
a vertical uplift or subsidence rate. Then, we define a constitutive law for each of these components (Carretier et al., 2016):

$$\epsilon_r = Kq^m S^n \text{ for river processes} \tag{2}$$

$$\epsilon_h = \kappa S \text{ for hillslope processes} \tag{3}$$

where $K$ [L$^{1-2m}$T$^{m-1}$], $\kappa$ [LT$^{-1}$] are erodibility parameters, $m$ and $n$ are lithology-dependent (different for bedrock or
sediment) erosion parameters, $S$ is the slope, $q$ [L$^3$T$^{-1}$] is the water discharge per stream unit width, corresponding to the
accumulation of the net precipitation rate (specified in the input or varying dynamically with elevation - Zavala et al. (2020))
from the highest to lowest cell according to a multiple flow algorithm, spreading the water discharge of the cell towards all the
lower cells among the eight neighbouring cells proportionally to their slope. The deposition rate is:

$$D_r = \frac{q_{sr}}{\zeta q} \text{ for river processes} \tag{4}$$

$$D_h = \frac{q_{sh}}{\frac{dx}{1-(S/S_c)^2}} \text{ for hillslope processes} \tag{5}$$

where $q_{sr}$ and $q_{sh}$ are the incoming river and hillslope sediment fluxes (total $q_s = q_{sr} + q_{sh}$) per unit width [L$^2$T$^{-1}$], $\zeta$ is a
river transport length parameter [T L$^{-1}$] and $S_c$ is a slope threshold. These fluxes are the sum of the sediment fluxes leaving
upstream neighbouring cells while the deposition rates on a cell are a fraction of the incoming sediment.


Equation 1 is solved using the forward finite volume method ($z$ represents the mean elevation of a cell). At each iteration,
cells of the model grid are ranked in a decreasing elevation order and then treated successively in that order to ensure that all
incoming water and sediment fluxes are known when treating a given cell. For each cell, incoming water fluxes are summed
with the local volume of precipitation falling on a cell over the time step. The resulting water flux is then spread among all
the downstream cells in proportion to their local topographic gradient (Multiple flow), or slope. Then the eroded flux is first
calculated on that cell from Equations 2, 3 using the steepest-descent slope among the downstream neighbouring cells. If
the erosion potential (erosion rate multiplied by the time step) is larger than the sediment volume present on that cell, the
bedrock is eroded in proportion to the remaining time step ((1 − sediment volume/erosion potential)). Then, the deposition
flux is calculated using Equations 4, 5. Erosion-deposition is first calculated for the hillslope processes and then for rivers.
Once the total eroded and deposited volumes are known, the elevation is modified based on the balance between these two vol-
umes. Sediments that leave the cell are spread downstream in proportion to the local slope of neighbouring cells. Then, the next
cell in the list is treated and so on. When all the cells have been treated, a new iteration begins and the time is incremented by $dt$.





Other processes such as lateral erosion, weathering and regolith development, orographic precipitation or the dynamic filling
of depressions are implemented but not described in this contribution because they are not useful in terms of presenting the
algorithm to calculate the CN concentrations in the grains (see Carretier et al., 2016, 2018; Zavala et al., 2020).

## 2.2   The grains in Cidre

Grains are clasts of any kind: e.g. a single mineral or a pebble comprised of multiple minerals, ranging in size from millimetres
to decimetres (Carretier et al., 2016). They are localized by the cell number where they are located (their precise 2D coordinates
within a cell are not known) and the depth of their centre beneath the Earth's surface. At the beginning of a simulation, their
number, radius $R$, location on the grid and depth are specified. For example, they can be set randomly on the grid and at depth,
or with a higher density in some regions, in order to simulate the different proportions of some minerals depending on the
underlying rock type.

Grains are moved once the erosion and deposition rates are known on the model grid at the end of a time step. They are
passive tracers, they do not influence the erosion and sediment calculated by Cidre and they are all independent from each
other. Grains are eroded, transported or deposited according to probabilities depending on the erosion and deposition rates. For
a grain on a cell, it is detached if the eroded layer on that time scale is deeper than or equal to the depth of the grain's bottom
$z_{\text{clast}}$. If the grain is not detached, its depth is updated according to the $dz$ calculated on that cell and for this time step. It is
possible for a grain to be detached but to not leave the cell to account for the time needed to travel a large cell, and for the
slower transport of a big grain. Carretier et al. (2016) found that to account for these effects and to predict the correct mean
sediment flux, the probability of leaving a cell should be $1.25\left(\frac{\epsilon dt}{2R}\right)\left(1 - \frac{z_{\text{clast}}}{\epsilon dt}\right)\delta$, where $\delta = 1$ if the direction of movement is
parallel to rows or columns, and $\delta = 1/\sqrt{2}$ along diagonals (a longer distance decreases the probability of leaving the cell).
The value of this probability is set to 1 if it exceeds unity, which may occur if the grain is at the surface and the erosion is larger
than the grain diameter. If the grain does not leave the cell, its depth is chosen randomly within the layer deposited on that cell
during the time step.

If a grain leaves the cell, it goes to one of the downstream neighbouring cells with a probability set as the ratio of the local
slope of the considered downstream cell and the sum of the downstream slopes. Once the grain has entered into one of these
downstream cells, the probability it will be deposited is the ratio of the sediment deposition flux in that cell and the sum of the
incoming sediment fluxes. If not deposited, the grain continues its travel and is exported towards one of the downstream cells,
and so on until it is deposited or it leaves the model grid. Then a new grain is treated and so on. When all the grains have been
treated, a new time step is initiated to calculate the local erosion and deposition on the model grid, etc.



## 3 Calculating the concentration for different CNs

### 3.1 CN evolution

The CN concentration $C$ at the centre of the grain varies based on

$$\frac{dC}{dt} = -\lambda C + P \tag{6}$$

where $\lambda$ is the radioactive decay rate [T$^{-1}$] and P is the CN production rate [$\mathcal{N}$/M/T] at any time $t$. The solution is calculated as (Carretier et al., 2009):

$$C(t) = e^{-\lambda t}(C_o + I) \tag{7}$$

Where $C_o$ is the initial CN concentration (see next section) and

$$I = \sum_n e^{\lambda t} P \, dt \tag{8}$$

where $n$ is the number of time steps $dt$ since the beginning of the landscape evolution simulation and $t = n.dt$

### 3.2 Production rate

The production rate depends on the nuclide considered. Here, we use a formulation based on three contributions by spallation (subscript "$sp$"), reactions induced by slow muon capture (subscript "$sm$") and by interactions with fast muons (subscript "$fm$") (Braucher et al., 2011).

$$P = (P_{sp} e^{-\rho z/\Lambda_{sp}} + P_{sm} e^{-\rho z/\Lambda_{sm}} + P_{fm} e^{-\rho z/\Lambda_{fm}}) \tag{9}$$

where $z$ is the depth of the grain's centre, $P_{sp}$, $P_{sm}$ and $P_{fm}$ are the production rates of a given CN at the Earth's surface by spallation, slow muon capture and fast muon interactions, respectively. $\rho$ is the grain density. $\Lambda_{sp}$, $\Lambda_{sm}$ and $\Lambda_{fm}$ [M L$^2$] are the respective attenuation factors with depth.

$$P_{sp} = P_{SLHL} f_{sp} S_{sp} \tag{10}$$
$$P_{sm} = P_{SLHL} f_{sm} S_{sm} \tag{11}$$
$$P_{fm} = P_{SLHL} f_{fm} S_{fm} \tag{12}$$

where $P_{SLHL}$ is the total sea-level/high-latitude production rate of the considered nuclide. $f_{sp}$, $f_{sm}$ and $f_{fm}$ are the fractions of this production rate due to spallation, slow muon capture and fast muon interactions. $S_{sp}$, $S_{sm}$, and $S_{fm}$ are the respective



scaling factors depending on latitude and elevation. In the simulations presented here, Stone (2000)'s model is used to calculate the scaling factors and fractions, but it is possible to implement Dunai (2000)'s model or the time-varying model by Lifton et al. (2014) accounting for variations in the magnetic field through geological timescales. Topographic shielding is not taken into account as the slopes in the following simulations are lower than $30^o$ (DiBiase, 2018).

The cosmogenic concentration is calculated for each grain at the end of a time step using Equations 7 and 8. For a grain in movement during the time step, the mean elevation and mean depth between its initial and final positions are used to calculate the CN production rate. For a grain leaving the grid, the $dt$ of this iteration in Equation 8 is decreased to account for the fact that the grain has spent only part of the time step in the grid: $dt$ is multiplied by the ratio between the depth of the grain on the starting cell of this time step and the eroded thickness on that cell during this time step.

Production rates for $^{10}$Be, $^{26}$Al, $^{21}$Ne and $^{14}$C are currently implemented in Cidre, but other nuclides can be easily added with specific production models (e.g. $^{36}$Cl, $^{3}$He, $^{78,80-82}$Kr - Dunai et al. (2022); Schaefer et al. (2022)).

### 3.3 Initial CN concentration

At the beginning of a simulation, an initial concentration $C_o$ must be specified for each grain. The choice of $C_o$ depends on the particular situation. For example, if we want to model the post-glacier evolution of a topography that was deeply eroded and previously protected from cosmic rays below a thick glacier, $C_o = 0$ at/g may be convenient. Another situation may correspond to a surface eroding at a constant rate $\epsilon$ [L/T] for hundreds of thousand years. In this case, a steady-state solution for $C_o$ is given by

$$C_o \quad = \quad P_{sp} e^{-\rho z/\Lambda_{sp}} * \frac{\Lambda_{sp}}{\rho\epsilon + \lambda\Lambda_{sp}} + P_{sm} e^{-\rho z/\Lambda_{sm}} * \frac{\Lambda_{sm}}{\rho\epsilon + \lambda\Lambda_{sm}} + P_{fm} e^{-\rho z/\Lambda_{fm}} * \frac{\Lambda_{fm}}{\rho\epsilon + \lambda\Lambda_{fm}} \tag{13}$$

The deeper the grains are set initially in the rock, the less the choice of $C_o$ matters for the following evolution because the production rate decreases exponentially with depth.

### 3.4 Grains revival

The fate of a grain is to leave the model grid at one of its outlets, where it then becomes a 'dead' grain. Every single grain may leave the model grid before the end of a simulation. For some applications, such as the study of the riverine detrital $^{10}$Be evolution at mountain outlets, the flux of grains reaching the outlet must be continuous. One approach is to populate the initial bedrock with a large number of grains at great depth so that there are always grains exhuming during the whole simulation. This is possible, but may require long computational times. One alternative is to reuse the grains leaving the model grid at each time step. We set them back to their initial cell, at a random depth between two specified values. Their initial concentration $C_o$ can be assumed to correspond to the steady-state value given by Equation 13 with the erosion rate corresponding to the erosion





rate on that cell calculated at the last time step. This approach allows us to handle a limited number of grains.

The other advantage of reviving grains at their initial location also deals with the theory of the [10]Be-derived catchment mean erosion rate, which we illustrate in the next section. This theory requires that the mean [10]Be concentration of grains at the outlet of a catchment reflects the ratio between the flux of [10]Be atoms and the flux of quartz (Brown et al., 1995). Because the revival of grains occurs more frequently where cells erode faster, there are proportionally more grains coming from these cells that reach the outlet compared to those coming from cells that are eroding more slowly. Statistically, this ensures that the mean

[10]Be concentration at the catchment outlet reflects the ratio between the [10]Be flux and quartz flux.

Using the steady-state $C_o$ when a grain is reintroduced in the grid is only an estimation of the true value that should integrate previous variations in the erosion rate, but if the depth at which the grains are set back is deep enough, this approach provides a good compromise between computing time and precision, as shown in the following examples. An appropriated revival depth

is below the attenuation length $\frac{\Lambda_{sp}}{\rho}$ of CN production by spallation. For a granitoid rock with a density of 2.7 g/cm$^2$, this attenuation length is approximately 65 cm. The production rate decreases exponentially by a factor of 2.7 between the surface and 65 cm. At one metre, the production rate is only roughly 21% of that at the surface. Deeper down (>3 m), the production by muons dominates but the production rate by muons adapts itself more slowly to variations in erosion rates (Braucher et al., 2003). Thus, setting back grains at depths deeper than 65 cm attenuates the error associated with the steady-state assumption

for defining $C_o$ if there are high frequency variations in the erosion rate. Conversely, the record of a previously higher erosion rate in the past can be lost in case of a topography that has evolved very slowly and which has undergone then a very strong decrease in erosion rate. Again, the deeper the grains are set back, the smaller the bias but also the smaller number of outgoing grains at each time step.

### 3.5    Pseudo code for the Cidre Erosion-Deposition and grain transport algorithm presented in this contribution

**Read** the input parameters, initial elevations and grains list

**While** the specified final time is not reached:

time = time + $dt$

Rank the cells in the order of decreasing elevation

**Do for each** cell in this order:

Calculate the slopes from the cell towards all the downstream directions

Calculate the water discharge by summing the incoming water flux and the local precipitation

Calculate the leaving water flux in each downstream cell direction in proportion to the local slope

(the water discharge $q.dx$ is now known on the grid)

**Do for each** cell in the same order:

Calculate the potential eroded volume of sediment by hillslope processes (Equation 2)

**If** the erosion is larger than the available volume of sediment on the cell:





Erode the bedrock but multiply the eroded volume by (1-sediment volume/potential erosion of sediment)

Calculate the potential eroded volume of sediment by river processes (Equation 2)

Calculate the deposited volume of sediment by hillslope processes (Equation 4)

Calculate the deposited volume of sediment by river processes (Equation 4)

Calculate the balance between incoming, deposited and eroded volumes and spread the leaving sediment volume among the downstream cells in proportion to their slope

Add the net elevation change from the difference between erosion and deposition (Equation 1)

Add the uplift to the elevation

**Do for each** grain in the grid until the grain is deposited or leaves the model grid:

**If** the grain does not move:

Update its depth

**Else If** it moves:

Draw the next cell among all downstream cells with a probability proportional to the slope

**If** the grain is not deposited on the next cell:

Continue to move the grain to the next cell

**If** the grain left the model grid:

**If** the revival option is true:

Set the grain back to its initial cell at a random depth between specified values

**Do for each** grain in the grid:

Calculate its CN concentration (Equations 7 to 10) in selected nuclides using the mean elevation and depth of its travel during the time step.

**Save** the results if the time fits the predetermined times.

## 4    Example: deriving the catchment mean concentration from 10Be in uplifting and down-wearing landscapes

### 4.1    Steady-state

We carry out a first test of the algorithm by comparing the true mean erosion rate calculated by Cidre (Equation 1) in a catchment with the $^{10}$Be-derived mean erosion rate inferred from the mean $^{10}$Be concentration of grains leaving the model grid at each time step. For this, we design a reference simulation using a grid of 100x100 cells with a size of 100 m (10 km$^2$), with only one imposed outlet in a corner fixed at an elevation of 0 m, and starting from a nearly horizontal topography with variations

in elevation obeying a Gaussian distribution with a standard deviation of 0.5 m. We impose a constant precipitation rate of 1 m/yr on the grid. The computation time step $dt$ is 100 years. The other model parameters are given in Table 1. We impose a first 10 Myr-period with a constant uplift rate of 1 mm/yr so that a dynamic equilibrium is reached, with a steady topography and a mean erosion rate on the grid equal to the uplift rate (Kooi and Beaumont, 1996). Then, we put 100,000 grains in the bedrock at a depth between 10 and 100 m with a $^{10}$Be concentration $C_o$ of zero at/g. These grains are progressively exhumed





during 0.1 Myr and then they leave the model grid. Once they leave the model, they are considered 'dead' for the rest of the simulation, we do not recycle them. We record the $^{10}$Be concentrations of the grains leaving the model grid at each time step and we calculate their mean $^{10}$Be concentration. In order to infer the mean erosion rate, we use the classic steady-state $^{10}$Be concentration model for a constant erosion rate given by (Lal, 1991; Braucher et al., 2011):

$$\bar{\epsilon} \quad = \quad \frac{1}{\rho \bar{C_o}} (\bar{P_{sp}} \Lambda_{sp} + \bar{P_{sm}} \Lambda_{sm} + \bar{P_{fm}} \Lambda_{fm}) \tag{14}$$

where $\bar{P_{sp}}$, $\bar{P_{sm}}$ and $\bar{P_{fm}}$ are the spatially averaged production rates over the model grid. As the CN production rates depend on elevation, we use the topography corresponding to the studied model time to calculate these values.

This model is based on Equation 13 but neglects the radioactive decay so that an analytical solution exists. At each time step, we calculate the $^{10}$Be-derived erosion rate if there are more than 10 grains that leave the model grid during the time step.
At the end of the simulation we can also calculate the average $^{10}$Be concentration of all the grains that went out during the 0.1 Myr steady-state period to calculate a mean $^{10}$Be-derived erosion rate that should equal the constant erosion rate of 1 mm/yr. We carry out other simulations with different uplift rates (0.01 mm/yr, 0.1 mm/yr, 0.5 mm/yr, 1.5 mm/yr) setting the initial depths of the grains so that there is the same number ($\sim 100$) of outgoing grains at each time step for all these simulations on average (e.g. between 10 and 160 m for $U = 1.5$ mm/yr).


Figure 1 shows the comparison between the true mean erosion rate (calculated from the landscape evolution model in Cidre) and the $^{10}$Be-derived erosion rate in each case. When averaged over the whole steady-state period, there is a good fit between the $^{10}$Be-derived and true erosion rates. The $^{10}$Be-derived erosion rate overestimates the true erosion rate for low erosion rates (0.01 mm/yr, 0.1 mm/yr) by about 10% because the radioactive decay is neglected in order to calculate the $^{10}$Be-derived
erosion rate, which is well known (Balco et al., 2008). From one time-step to another, there is a variability that depends on the number of outgoing grains as well as on the magnitude of local erosion on the grid. The variability is higher for larger erosion rates ($\pm 20\%$ for $U = 1.5$ mm/yr), which is consistent with erosion that is dominated by hillslope processes with slopes near the critical slope $S_c$ (Equations 4). At each time step, a thick layer of tens of centimetres can be removed, including one grain on average in each cell, at different depths depending on the cell, and thus with very different $^{10}$Be concentrations.

**4.2  Transient erosion rate**

Here, we test the consistency between the true and $^{10}$Be-derived erosion rates during the transient stage of the topography adaptation to uplift. We use the same parameters as in the reference simulation described in the previous section. We impose a first 5 Myr-period with a constant uplift rate of 1 mm/yr and a second 5 Myr-period with a constant uplift rate of 0.1 mm/yr that is ten times lower. The (true) evolution of the mean erosion rate is composed of a transient period and then a dynamic
equilibrium where $\epsilon = U$ on each cell (Figure 2A). During the first period, the transient period begins with the establishment of a drainage network and then an increase in the slopes leading to an increase in the mean erosion rate with a classic convex



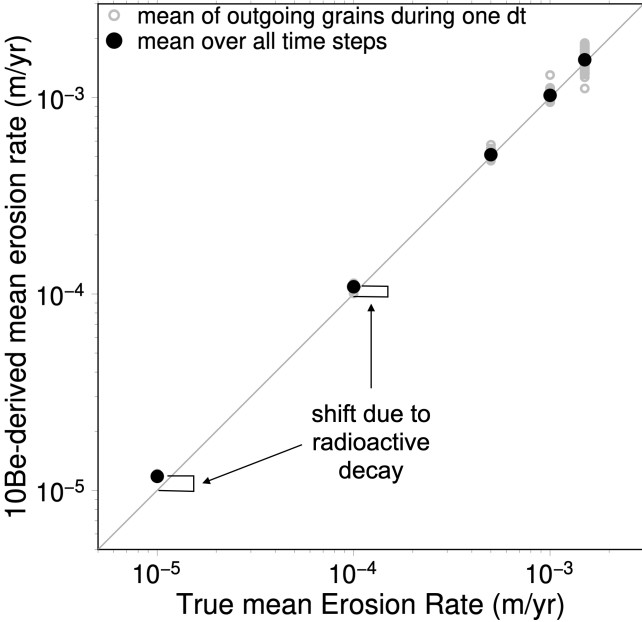

**Figure 1.** Comparison between the true (calculated from the landscape evolution model in Cidre) and $^{10}$Be-derived erosion rates at dynamic equilibrium ($U = \bar{\epsilon}$) for different values of uplift rates (i.e. different simulations). The overestimation (shift) of the $^{10}$Be-derived erosion rates for a low erosion rate ($10^{-5}$ and $10^{-4}$ m/yr) comes from the absence of radioactive decay in Equation 14 used to infer the $^{10}$Be-derived erosion rates whereas radioactive decay is taken into account in Cidre. Radioactive decay slightly decreases the mean $^{10}$Be concentration calculated by Cidre, and thus the apparent inferred erosion rate neglecting radioactive decay, which is inversely proportional to the $^{10}$Be concentration, is slightly overestimated.

curve, before reaching the dynamic equilibrium (Bonnet and Crave, 2003; Carretier et al., 2009) (Figure 2A). The maximum dynamic equilibrium elevation reaches 1800 m. In the second period, the mean erosion rate decreases to the new dynamic equilibrium value with a maximum elevation of 340 m.


At the beginning of the simulation, we spread 10,000 grains of quartz with a radius of 1 mm (1 per cell on average) at a randomly chosen location and depth, where the depth is between 3 and 6 m. Their initial $C_o = 0$ at/g. Contrary to previous simulations, where grains were dead once they left the model grid, we recycle them each time they leave the model grid. The grains are exhumed and transported, and when they leave the model grid, they are set back to their initial cell at random depths between 1 and 2 m. A grain is set back with a $^{10}$Be concentration $C_o$ corresponding to the steady-state concentration (Equation 13) calculated using the current erosion rate of the cell where the grain is set back. The minimum depth of 1 m ensures that most of the $^{10}$Be acquisition (79%). This depth will be evaluated later.



**Figure 2.** Reference simulation. A- Evolution of the mean erosion rate evolution with two periods of constant uplift rates. B- Snapshot of three stages in the topographic evolution. The number of displayed grains is divided by 10 for easier viewing.



| | |
|---|---:|
| $\kappa$ | $10^{-4}$ m/yr (equal for both sediment and bedrock) |
| $S_c$ | 0.83 m/m (equal for both sediment and bedrock) |
| $K$ | $10^{-4}$ yr$^{-0.5}$ (equal for both sediment and bedrock) |
| $\zeta$ | 1 |
| $\underline{U}$ | $10^{-3}$ m/yr |
| cell size | 100 m |
| No. of rows x columns | 100 x 100 |
| time step $dt$ | 100 yr |
| No. of grains | 10000 |
| Revival depth | random between 1 and 2 m |
| $P_{SLHL}$ | 4 atoms/g/yr |
| $\Lambda_{sp}$ | 150 g/cm$^2$ |
| $\Lambda_{sm}$ | 1500 g/cm$^2$ |
| $\Lambda_{fm}$ | 4320 g/cm$^2$ |
| $f_{sp}$ | 0.9886 |
| $f_{sm}$ | 0.0027 |
| $f_{fm}$ | 0.0087 |

**Table 1.** Model parameters used in the reference simulation. The underlined parameters are those that are varied in the other simulations.

Figure 2B shows three snapshots of the topography and grains at 0.35, 5 and 10 Myr. The number of displayed grains is
divided by 10 for easier viewing. In figure 2B, the large cubes show grains that are in movement during the last time step,
whereas the smaller ones show grains still in the bedrock at depth. The first snapshot corresponds to the period of drainage
network growth (see Figure 2A). Not all cells are connected to the imposed outlet, and therefore, the divide of the associated
catchment is expanding towards the boundary of the model grid. The 5 Myr step is the topography at dynamic equilibrium
with a homogeneous and constant erosion rate of 1 mm/yr. The 10 Myr snapshot corresponds to the dynamic equilibrium of
the second period with a final mean denudation rate of 0.1 mm/yr, and consequently a lower elevation (see Figure 2A).

The number of grains leaving the model grid at each time step increases through time in the first period (Figure 3A) because
the mean erosion rate increases (Figure 3B). During the second period, the mean erosion rate decreases and so does the number
of grains leaving the model grid (Figure 3). The mean $^{10}$Be concentration increases in the first hundreds of thousand years
because, during the progressive establishment of the drainage network, many grains have stayed in the former surface for a
long time before being exported out of the catchment. During the uplift of this surface, the $^{10}$Be production rate increases. The
long residence time and increasing $^{10}$Be production rate have produced grains with high $^{10}$Be concentrations, which explains
the increase in the mean $^{10}$Be concentration during network growth. Once the drainage network is completely established, the
$^{10}$Be concentration decreases slightly and stabilizes to a nearly constant value although the erosion rate is increasing rapidly:
the increase in the $^{10}$Be production rate due to the increasing elevation is compensated by a decrease in the residence time





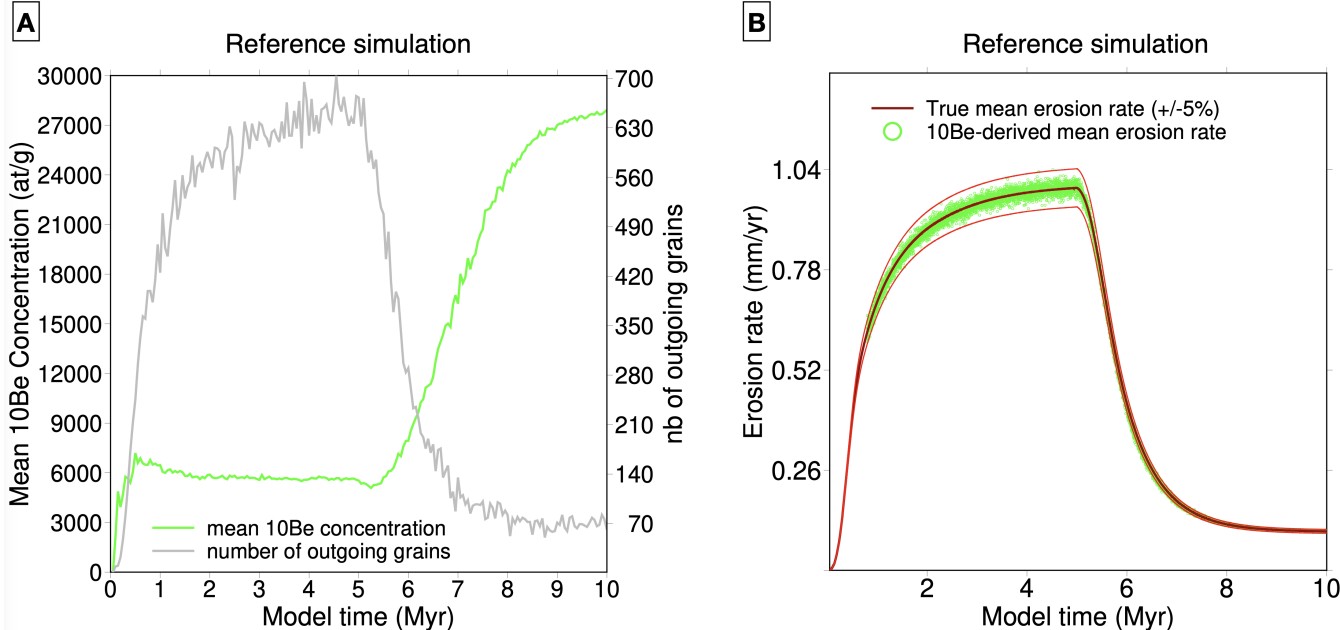

**Figure 3.** Reference simulation. A- Evolution of the number of outgoing grains during a time step (100 yrs) and the mean $^{10}$Be concentration averaged over the outgoing grains. B- The true mean erosion rate calculated by Cidre and the mean $^{10}$Be-derived erosion rate calculated from the mean $^{10}$Be concentration of outgoing grains.

of the grains as the erosion rate increases. This evolution illustrates that a record showing a constant $^{10}$Be concentration may not be a diagnostic for a constant erosion rate. During the second period, the mean concentration in $^{10}$Be increases because, although the elevation and thus the $^{10}$Be production rate decrease, the $^{10}$Be mean concentration is dominated by the longer residence time of grains following the decrease in erosion rate.


Figure 3B shows that the mean $^{10}$Be concentration of grains leaving the outlet does not depend on the number of grains, and high frequency variations in this number from one time step to another generate smaller variations in the mean $^{10}$Be concentration. The continuous and smooth evolution of the mean $^{10}$Be concentration shows that the outgoing grains correctly sample the model grid although they are 14 to 100 times less numerous than the model cells, even during the transient periods where 305 the erosion rate is heterogeneous.

We now use the mean $^{10}$Be concentration $\bar{C}$ of the grains leaving the catchment during a time step to calculate the mean catchment erosion rate $\bar{\epsilon}$ through time using Equation 14. $\bar{\epsilon}$ is not calculated if the number of grains is smaller than 20. Furthermore, $\bar{\epsilon}$ is not calculated during the development of the drainage network for practical reasons as the catchment is smaller 310 than the whole model grid. Each $^{10}$Be-derived $\bar{\epsilon}$ value corresponds to the average of outgoing grains over a time step of 100 years. Figure 3C shows the good match between the true mean erosion rate and $\bar{\epsilon}$ through time. The variation of $^{10}$Be-derived




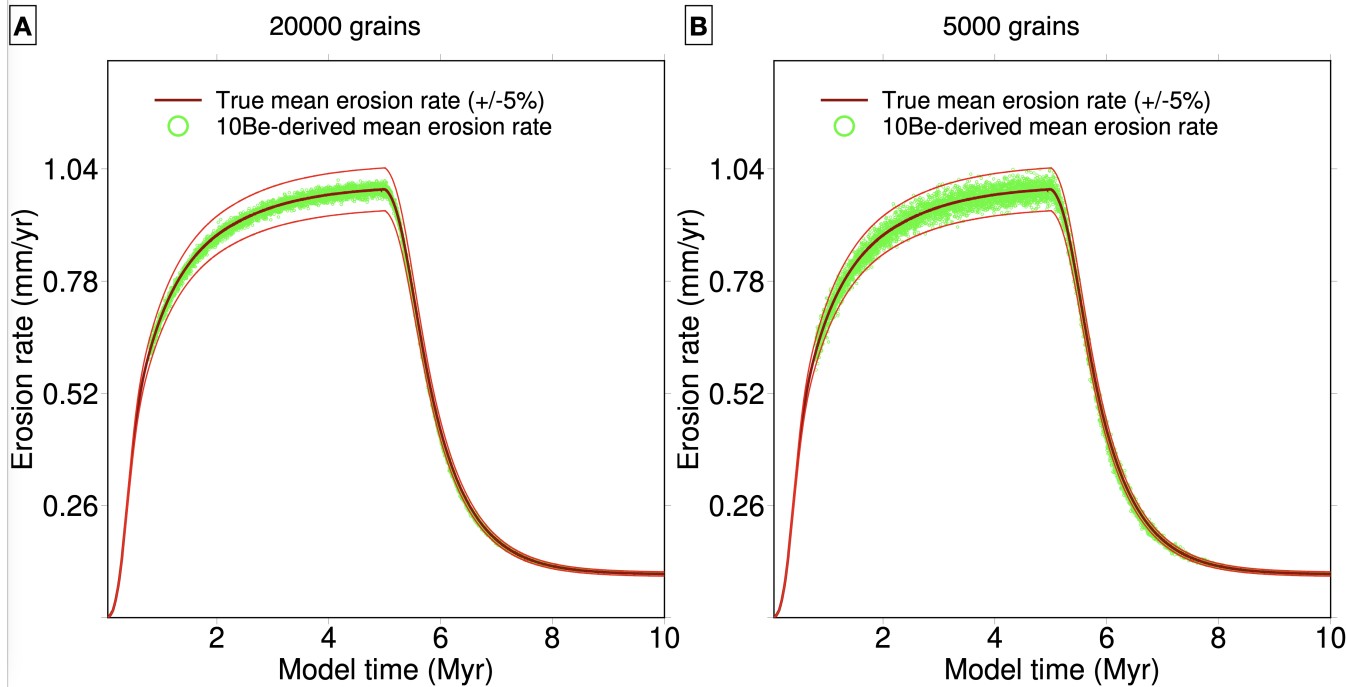

**Figure 4.** Effect of seeding the model with a different number of grains on the mean $^{10}$Be-derived erosion rate calculated from the mean $^{10}$Be concentration of outgoing grains. A- 20,000 grains (2 per cell in average). B- 5,000 grains (1 out of every 2 cells on average).

$\bar{\epsilon}$ is less than 5% around the true value for the two periods.

Doubling the initial number of grains decreases the scattering to 1% around the true value while halving the number of
grains increases it to 5% (Figure 4).

Decreasing the time step to 20 years increases the scattering to 5% because there are fewer grains leaving the model grid during a time step (Figure 5). There is also a very slight ($\sim$ 1%) overestimation of the mean erosion rate on average when it becomes larger than about 0.8 mm/yr because a small time step increases the probability for a grain to be temporarily stored in
a cell. Grains coming from cells far from the outlet may take several time steps to reach the outlet once they are detached from the bedrock. A grain close to the outlet may take less time. Consequently, there are proportionally more grains coming from cells close to the outlet. As these grains are located at lower elevations, with smaller $^{10}$Be production rates and thus smaller $^{10}$Be concentrations, the mean $^{10}$Be concentration of the outgoing grains is slightly underestimated. In turn, the $^{10}$Be-derived erosion rate is slightly overestimated.





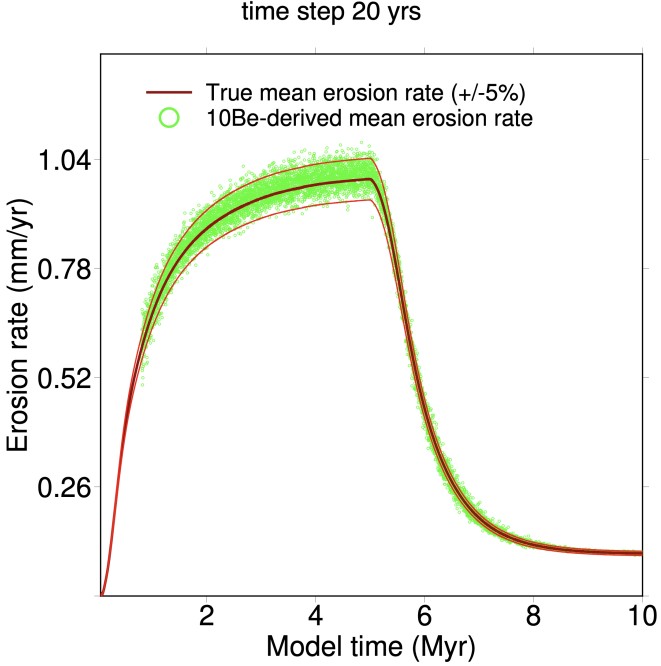

**Figure 5.** Effect of dividing the calculation time step by 10 on the mean $^{10}$Be-derived erosion rate calculated from the mean $^{10}$Be concentration of outgoing grains. The variability of the mean $^{10}$Be-derived erosion rate increases because there is a smaller number of outgoing grains over a smaller time step.

Decreasing the cell size to 25 m and the time step to 50 years does not change the goodness of fit between the $^{10}$Be-derived and true $\bar{\epsilon}$ (Figure 6).

In order to test the effect of catchment size on the calculation of $^{10}$Be-derived $\bar{\epsilon}$, we use a model grid with 200x200 cells, 330 which is four times larger than the reference simulation, but we leave the other parameters as in the reference simulation. We seed the model grid with 40,000 grains, i.e. one per cell on average, as in the reference simulation. Figure 7 shows the good match between the $^{10}$Be-derived $\bar{\epsilon}$ and the true value in this case too. The smaller variability of the $^{10}$Be-derived $\bar{\epsilon}$ around the true value is due to the fact that there are four times more grains that leave the grid at each time step.

We now rerun the reference simulation but divide the uplift rate by ten in the two periods. Figure 8A shows that the $^{10}$Be-335 derived $\bar{\epsilon}$ is overestimated by approximately 2% during the dynamic equilibrium of the first period (4-5 Myr). This is due to the model in Equation 14 used to calculate $\bar{\epsilon}$ that neglects the radioactive decay. Here, radioactive decay influences the $^{10}$Be concentration because the clast residence time in the bedrock is longer for small uplift rates (Lal, 1991). In the second period, the $^{10}$Be-derived erosion rate is overestimated by roughly 20% during the transient adjustment to a smaller uplift rate (6-8 340 Myr). This overestimation comes from the delayed response of CN for a low erosion rate. The response time to adjust to a new




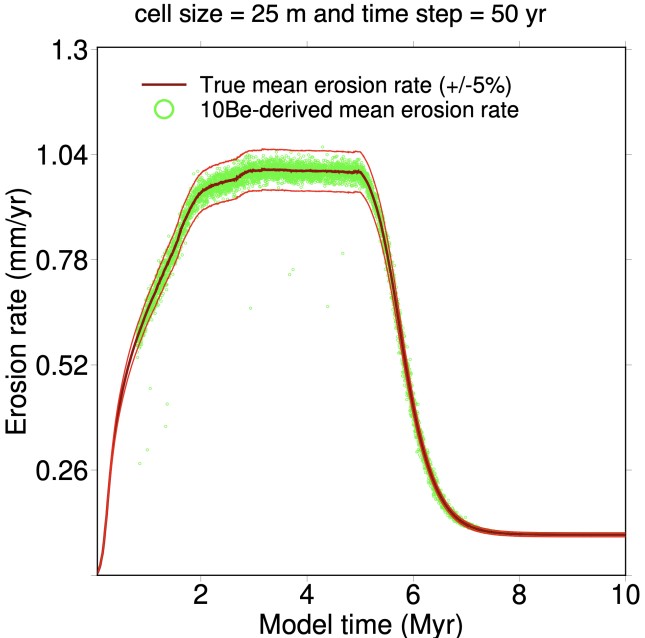

**Figure 6.** Effect of decreasing the cell size to 25 m (keeping the same number of cells) on the mean $^{10}$Be-derived erosion rate calculated from the mean $^{10}$Be concentration of outgoing grains. The different shape of the denudation curve compared to the reference simulation in Figure 3 comes from the increase in slopes in this simulation due to the smaller cell size, leading to a larger contribution of hillslope processes when the slope approaches the critical slope $S_c$ in Equation 4.

erosion rate $\epsilon$ is about four times $1/(\lambda + \frac{\rho}{\Lambda_{sp}}\epsilon)$ when only considering CN produced by spallation (Lal, 1991). With $\epsilon$ between 0.1 and 0.01 mm/yr, the response time is between 60,000 yr and 600,000 yr. Consequently, the CN concentration is out of phase during the rapid transient decrease in the erosion rate for the second period. This delay was not observed in the reference simulation because the response times were ten times shorter. Furthermore, the variability of $^{10}$Be-derived $\bar{\epsilon}$ tends to increase

for smaller uplift rates because the number of outgoing grains is smaller at each time step. In this simulation, the number of outgoing grains varies between 100 and 10. It is remarkable that with such a small number of grains (particularly during the second period with a very low uplift rate), the $^{10}$Be-derived $\bar{\epsilon}$ remains a good estimate of the true $\bar{\epsilon}$. When the number of grains is multiplied by four, this decreases the variability (Figure 8B).

As there is a rapid decrease in the erosion rate, but a long residence time of grains in the last metres because the erosion rates are low in these last simulations, one may question the chosen revival depths of the grains because they may be too shallow to record the previous decrease in erosion. We test this by changing the revival depth of the grains to between 4 and 6 m. Figure 8C shows that the delay in the $^{10}$Be-derived erosion rate between 6 and 8 Myr is similar, with a larger variability due to the smaller number of outgoing grains at each model time.




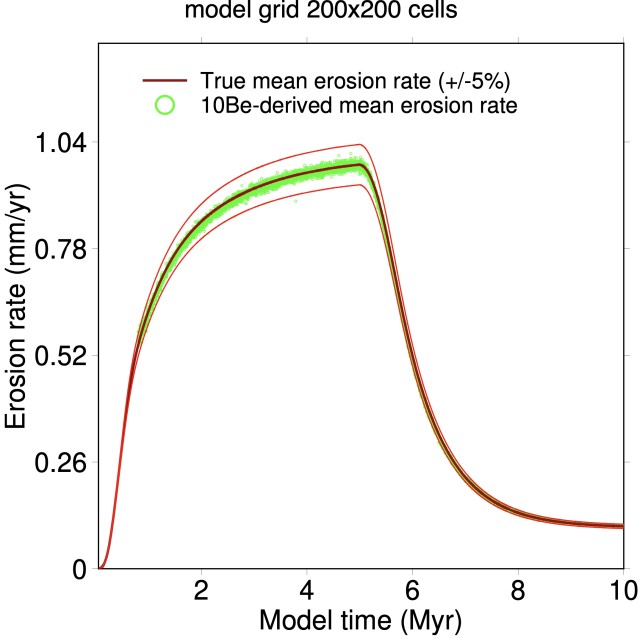

**Figure 7.** Effect of multiplying the model grid size by four on the mean [10]Be-derived erosion rate calculated from the mean [10]Be concentration of outgoing grains. The number of grains is also multiplied by four to have one grain per cell on average as in the reference simulation.

In order to further test if a revival depth between 1 and 2 m gives good results, we carried out three final simulations using the parameters of the reference simulation, but imposing a constant uplift rate of 0.1 mm/yr and an oscillatory precipitation rate between 0.5 and 1 m/yr during 10 Myr. The three simulations correspond to three different periods of oscillation constituting the Milankovitch cycles: 23 kyr, 41 kyrs and 100 kyrs. It is predicted that the [10]Be-derived erosion rate should be shifted with a lag that increases with the period of oscillation (Schaller and Ehlers, 2006). Figure 9 illustrates several cycles in the last 200 kyrs of these simulations. It shows that the lag between the [10]Be-derived and true erosion rate signals increases with the oscillation period. The lags are very similar to the ones found by Schaller and Ehlers (2006) in the case of one-point source simulations for the same oscillation periods, magnitude and true erosion rates (see Figures 5d, 5e and 5f in Schaller and Ehlers (2006)).

## 5 Discussion and conclusion

### 5.1 Limitations and advantages

The main limitation of this approach deals with the computational time when the number of grains is large. Some study cases may require a huge number of grains. In practice, the feasibility of these simulations will depend on the computation facilities. To give an indication, the reference simulation with 100x100 cells and 10,000 grains runs in half an hour on a personal





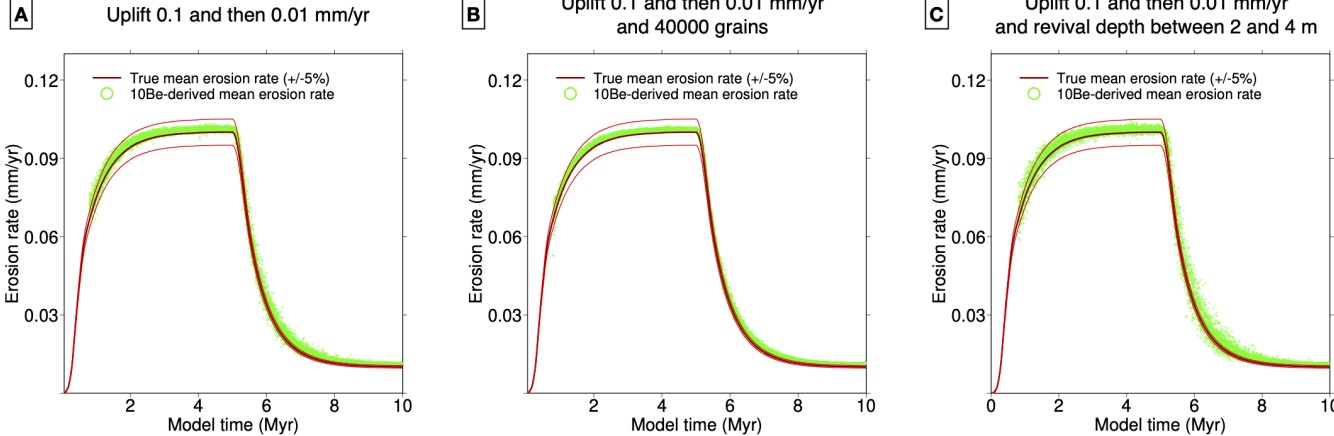

**Figure 8.** Effect of an uplift rate that is 10 times smaller. A- The number of grains is 10,000 as in the reference simulation. B- The number of grains is 40,000. C- The number of grains is 10,000 but the (revival) depth at which the grains are set back to the model grid is between 2 and 4 m instead of 1 and 2 m. In the three cases, the slight 1-2% overestimation of the $^{10}$Be-derived erosion rate during the first dynamic equilibrium stage (4-5 Myr) comes from the model used to derive erosion rates that neglect the $^{10}$Be radioactive decay, which only matters for low erosion rates. During the second period (6-8 Myr), there is a significant overestimation of the true erosion by approximately 20%. This overestimation comes mainly from the delayed response of the CN concentration for a low erosion rate. In C, the variability is larger because there are fewer grains going out at each model time step.

computer. The simulation with a grid and number of grains four times larger runs in 2.5 hours. Parallelism of the grains'
calculation is straightforward as they are independent of each other, and would decrease the running time. As the precision
of the results depends on the number of grains, the model can be adapted according to the studied process. For example, a
higher density of grains is required to study the CN detrital signal associated with climate variations for the 20 kyr period
compared to the density required for variations in the 100 kyr period. It is possible to determine a simple quality control of the
results by setting a minimum number of grains leaving the model grid to interpret the CN concentrations. Beyond the number
of grains, the choice of time step and revival depths have an influence on the resulting cosmogenic concentrations, but the
simulations presented here show that these choices do not imply variations in inferred erosion rates larger than 10% around
the true value (when the CN and the erosion rate signals are in phase). This error remains smaller than or equal to the uncer-
tainty of CN-derived erosion rate in most cases, given the external uncertainties on the CN production rates (Balco et al., 2008).

  In the current implementation, grains do not physically erode during their transport (but they can weather and decrease in
size chemically - Carretier et al. (2018)). In the real world, attrition can significantly change the CN concentration within and
at the surface of decimetric pebbles. In the model, this would require tracking the CN concentration of the layers within the
pebbles. This was done in Carretier and Regard (2011)'s model, which served as a basis for the grains model in Cidre; there-



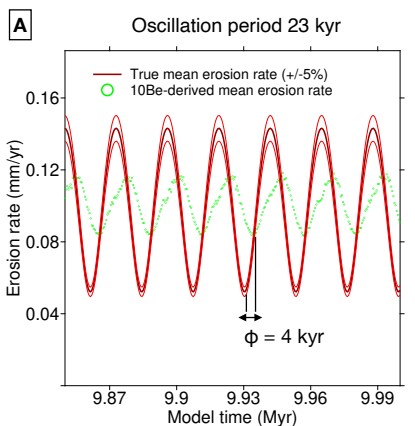 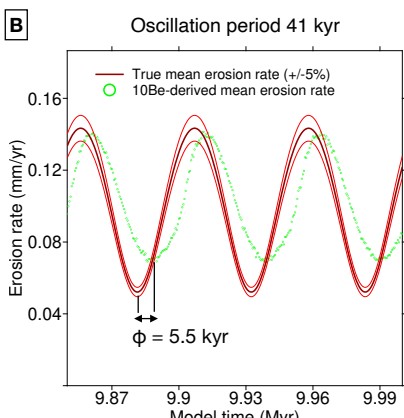 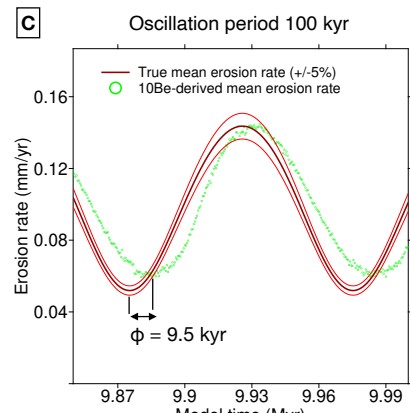

**Figure 9.** Simulations using the parameters of the reference simulation, but a constant uplift rate of 0.1 mm/yr and oscillations of the precipitation rates between 0.5 and 1 m/yr with different periods. A- 23 kyr. B- 41 kyr. C- 100 kyr. The observed lags $\phi$ are similar to the ones calculated by Schaller and Ehlers (2006) for similar conditions but for a theoretical one-point source catchment. This consistency shows that the revival procedures of grains at depths between 1 and 2 m give robust results.



fore, that it would be straightforward to implement it in Cidre as well.

      Modelling deep landslides (Campforts et al., 2020) would be a limitation to the grain revival process. If the specified maximum depth at which a grain is set back to the grid is much smaller than the thickness of the landslides, there is a risk that the mean CN concentration of grains in the landslide overestimates the mean value of the landslide. In such cases, the maximum

revival depth of the grains must be adequately adapted to be greater than the maximum landslide thickness.

      The revival of dead grains is a useful approach for the statistics of CN at the catchment outlet, but this approach is an approximation. It is fundamental that the depth at which the grains are set back to the model grid is much larger than the attenuation length of production by spallation ($\frac{\Lambda_{sp}}{\rho} \sim 65$ cm in granitic rocks). If not, there is a risk that the revival CN concentration $C_o$ of

the grain is a very poor approximation of the erosion history of its cell if the erosion rate has decreased very quickly and with a huge magnitude. Figure 9 shows that setting back the outgoing clasts at depths larger than 1 m with the $^{10}$Be concentration corresponding to the equilibrium value with the local erosion rate gives robust results.

      In a Lagrangian formulation, the approach by discrete grains has advantages. The alternative Eulerian approach would

require calculating and tracking the concentration of CNs at different depths below each cell at each time step (Niemi et al., 2005). Petit et al. (2023) present a simplified approach implemented in the LEM Badlands model based on the computation of the CN concentration at the Earth's surface only without calculating the CN concentration profile at depth. This level of simplification is required in this case for practical reasons of computation in a LEM. Tracking the whole depth profile evolution in the whole model grid cell where erosion and deposition can alternate at each iteration would be prohibitive in

terms of computational time and would be difficult to implement (Petit et al., 2023). One advantage of the Lagrangian approach is that it can tackle more complex erosion-deposition scenarios compared to the Eulerian approach (Knudsen et al., 2019). For example, it is simpler to calculate the evolution of a given grain experiencing a complex erosion-deposition history of the surface above it rather than the evolution of a (deep) depth profile of the concentration for each time step below a surface that is constantly changing in terms of elevation. From the grain's point of view, it is easy to calculate the evolution of CN in the grain

when it is stored, buried and eroded again stochastically, or when the soil above it is alternatively eroded or buried, as only the grain's depth has to be adapted. Note that the difference in density between rock and sediment is not taken into account when calculating the CN production rate in the simulations presented in this work, but it is possible to simply implement this. The Eulerian approach requires tracking the sources, and averaging the concentration from these different sources in cells during the transport of sediment, assuming perfect mixing. This process loses some information about the distribution of CN

for a population of grains. On the contrary, the distribution of the CN concentration is fully conserved by treating the grains separately. One drawback to the grain-by-grain approach is that the average CN concentration of the transported sediment is not precisely known, with an uncertainty that decreases with the number of grains. On the other hand, when the full distribution of the CN concentrations of a population of grains is known, then it becomes possible to study part of the stochasticity of erosion-deposition processes on hillslopes and in rivers, including the long-term temporary storage of some grains that may



contribute significantly to the CN concentration average (Carretier et al., 2019). The potential applications of this are described below.

## 5.2   Potential applications

The method using CN in riverine sediment to quantify modern and palaeo mean catchment denudation rates assumes that the CN concentrations of any grain have been entirely acquired on the hillslopes. Yet, the temporary storage and recycling of
sediment grains between the eroding sources and sedimentary basins may change their CN concentration depending on the considered nuclide or the climatic context (Wittmann and VonBlanckenburg, 2009; Carretier et al., 2019). Storage and recycling can delay or obscure the transmission of an erosion signal from source to sink (Carretier et al., 2020; Tofelde et al., 2021). Detrital signals in the CN concentrations can also be used positively to study the erosion-deposition processes on hillslopes (Slosson et al., 2022), burial histories in basins (Balco et al., 2013; Sanchez et al., 2021), the distribution of residence times
and transport lengths in rivers (Carretier et al., 2019) or to infer the palaeo-denudation rates of mountains (Charreau et al., 2011). Nevertheless, linking a CN detrital signal with landscape evolution is not straightforward and still faces the difficulty of modelling stochastic processes in a landscape evolution model. A forward model such as Cidre, which simulates CN in distinct grains that move stochastically, would help. For example, the conclusions of several studies on palaeo-denudation rates over the Plio-Pleistocene periods suggest either constant or modest variations in the denudation rate in Asia or in the Andes
(Charreau et al., 2011; Madella et al., 2018; Lenard et al., 2020; Charreau et al., 2021), or a significant change recorded in the last glacial cycle in river terraces or at the outlet of small mountainous catchments (Schaller et al., 2002; Mariotti et al., 2021). Some factors that should be explored numerically with a numerical model such as Cidre to better understand the CN detrital signals (Petit et al., 2023) include: the effect of catchment size, uplift rate or grain size, whether a flood plain is present or not, the frequency of climatic variations, variations in the elevation of the eroded sources.


Cidre includes the possibility to produce a regolith corresponding to the weathering of the underlying bedrock (Carretier et al., 2014, 2018). Once a grain is located in the regolith during its exhumation on the hillslopes, it continues to be exhumed towards the surface until it is detached. Alternatively, its depth could be chosen randomly at each time step to simulate bioturbation or physical creep within the soil. This option is easy to implement and would make it possible analyse the effect of
different soil processes on the riverine detrital CN signal within the framework of reservoir theories (Mudd and Yoo, 2010).

One of the advantages of modelling grains is the simplicity to track the evolution of distinctive cosmogenic nuclides, taking advantage of their different radioactive decay rates. When grains experience complex temporary burial and recycling histories, their initial concentration ratio on hillslopes varies downstream. It would be also quite simple to implement the evolution of the
meteoritic $^{10}$Be concentration at the surface of the grains, or their optically stimulated luminescence (OSL) dosimetry based on Guyez et al. (2023). The distributions of various CN concentrations in a river sample, e.g. $^{10}$Be, $^{26}$Al and $^{14}$C, contain information about possible past changes in the mean catchment erosion rates and residence times in fluvial systems (Repasch et al., 2020; Ben-Israel et al., 2022). We need to establish a link between erosion rate changes and the detrital CN concentrations

in different nuclides and potentially other properties such as OSL dosimetry. Doing so may allow us to address recent changes
over the last few centuries associated with natural and anthropic modifications of the landscape.

## 6 Conclusion

We present a development of the Cidre model by coupling landscape evolution and the evolution of CN concentrations in
distinct grains. The algorithm is tested by deriving the mean catchment erosion rate from the $^{10}$Be concentration of grains
leaving an uplifting catchment. The main limitation is the number of grains set in a simulation to achieve the desired precision.
This Lagrangian approach allows to fill the gap that exists between landscape modelling, which is used to help understand
variations in the elevation and erosion of landscapes, and field data, which often correspond to the CN concentrations of grains
in a soil or river sample.

*Code availability.* The Cidre source codes are available here https://gitlab.com/geomorphotoulouse/cidre under the opensource CeCILL v2.1
licence. The code is also permanently deposited on the HAL repository with the number hal-04141239v1 (https://hal.science/hal-04141239).

*Author contributions.* S. Carretier and V. Regard designed the study and the cosmogenic nuclide model, S. Carretier and Y. Abdelhafiz
implemented the cosmogenic nuclide model with the help of B. Plazolles. S. Carretier wrote the paper with inputs of all the co-authors.

*Competing interests.* No competing interests.

*Acknowledgements.* We would like to thank Marc De Rafelis for useful discussions. This research was supported, in part, by the French
ANR, and the WIVA, PANTERA and WEARING-DOWN projects.



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
