# Peer review of "Modelling detrital cosmogenic nuclide concentrations during landscape evolution in CIDRE V2.0"

_EGUsphere, 2023_

## Referee Comment (RC1)

Line 3:" A model could help…": awkward sentence. I suggest something like "to address this, a model can be designed to explore the statistical properties of CN concentrations in sediment grains."

Line 59: Why hillslope erosion is not simulated by a diffusion equation? Equation 3 should look like:

$$\epsilon_h = \nabla \cdot (\kappa \nabla h)$$

I don't understand how drainage divides get eroded with equations 2 and 3 since basically erosion is null when the slope is null. Am I missing something?

Line 75: You use MFD to distribute the incoming water flux from the donor node to the receiver nodes, but then you only use the "steepest-descent slope" when you compute the erosion potential of the donor node. Somehow it means that the water that is given to the other nodes does not contribute to erosion. Why not computing the erosion potential of a donor node as the sum of the contribution of each receiver node proportionally to their slope?

Line 81: "Sediments that leave the cell are spread downstream". Do you mean "distributed to downstream cells"?

Line 88: Composed instead of comprised

Line 89: "They are localized by the cell number where they are located": not clear to me. Do you mean they have an index corresponding to the cell number where they were initially sown?

Line 98: "For a grain on a cell, it is detached if the eroded layer on that **time step** is **thicker** than or equal".

Line 110: Sediment deposition volume instead of flux; I would keep the term "flux" for something that is moving.

Line 197 and 206: "Outgoing water flux" instead of "leaving water flux"

Line 202:" Erode the bedrock but multiply the eroded volume by (1-sediment volume/potential erosion of sediment)" Awkward sentence. Could you rephrase it please? If the volume to be eroded is greater than the volume of sediment available, the bedrock is eroded by the remaining quantity. Is it correct?

Line 214: what do you mean by "draw the next cell"?

Lines 215-216: Shouldn't this be in a while loop (with lines 213-214)?

Line 223: Do you mean user-determined output times?

Line 299: How do you define the "residence" time? This term is not clear to me. I would say that the "residence time" of sediments is the time spent in a given system (river network for instance) whatever they are exposed to cosmic rays or not. On the other hand, the exposure time should be the duration during which the sediments are exposed, even partially, to cosmic rays whatever the system they reside in. Could you please be more specific on what you call the residence time?

Line 301: Figure 3A instead of 3B

Line 322: Having seen this, I think it would be interesting to see the effect of a variable Quartz source distribution (i.e., more abundant in the upstream or downstream parts of the catchment for instance) on the resulting 10Be-derived erosion rate. Maybe add this topic in the Discussion section also?

Line 362: You do not discuss the amplitude. It seems that the amplitude is largely underestimated for the short period oscillations. Why is it so?

---

## Referee Comment (RC2)

Revisiting the grammar throughout the manuscript would do the work better justice but the paper is well structured and easy to follow. I have highlighted some of the sentences below that would benefit from clarification/rewording. The conclusion would also be made more impactful if there was a more holistic summary of the work. Minor comments below:

Line2: remove 'the' from 'the relief evolution'

Line3: suggest change to 'Models can be used to explore the statistics of CN concentrations in sediment grains'

Line7: change to 'The concentrations of various CNs can be tracked in these grains.'

Line10: not clear what a 'grain-by-grain distribution' is. Rephrase sentence?

Line12: Rephrase, e.g. 'We illustrate the robustness and limitations of this approach by deriving the catchment-average erosion rates from the mean 10Be concentration of grains leaving a synthetic catchment, and comparing them to the erosion rates calculated from sediment flux, for different uplift scenarios.'

Line33: 'but without taking the evolution of the relief into account.' Could you specify why this is important?

Lines47, 226, 261 etc: I think the clarity of the manuscript could be improved by better defining what is meant by the 'true rate' and using this term consistently throughout the manuscript.

Line68: What slope threshold and transport length do you use?

Line85: Rephrase 'they are not useful in terms of presenting the algorithm to calculate the CN concentrations in the grains'.

Line92: Rephrase: 'For example, they can be set randomly on the grid and at depth, or with a higher density in some regions, in order to simulate the different proportions of some minerals depending on the underlying rock type.'

Line189: I like the pseudo code!

Line268: Clarify: 'In the second period, the mean erosion rate decreases to the new dynamic equilibrium value with a maximum elevation of 340 m.'

Line273: different wording? 'where grains were dead…'

Figure 1 caption: Clarify: 'Radioactive decay slightly decreases the mean 10Be concentration calculated by Cidre, and thus the apparent inferred erosion rate neglecting radioactive decay, which is inversely proportional to the 10Be concentration, is slightly overestimated.'

Line297: interesting!

Line329: Why did you chose to test this variable? Include a sentence earlier in the manuscript e.g. paragraph starting line45.

Line348: 'When the number of grains is multiplied by four, this decreases the variability (Figure 8B).' Could you expand on the significance of this? Perhaps in the discussion.

Line399: rephrase 'In a Lagrangian formulation, the approach by discrete grains has advantages.'

Line433: rephrase 'and still faces the difficulty of modelling stochastic processes in a landscape evolution model'

Line437-439: Could this be expanded on a little? I think it is an interesting part of the discussion. Could you also look at connectivity?

Line456: Reword?: 'We present a new coupling of landscape evolution model Cidre with a model of CN concentrations in individual grains.

Line458: Clarify: 'The algorithm is tested by deriving the mean catchment erosion rate from the 10Be concentration of grains leaving an uplifting catchment.' – how does this test the algorithm?

---

## Author Comment (AC1)

The approach used here consists of coupling a Lanscape Evolution Model (LEM) developed by the 1st author with well-known cosmogenic nuclide (CN) production and decay laws, in order to track individual particles (grains) journey from their source to their sink (here, when they leave the model grid). This allows the authors to evaluate statistically how the 10Be signal carried by a population of grains in riverine sands is representative of the average denudation rate of the upstream catchment. For now, I guess the aim of the authors is to demonstrate the model's ability to achieve the desired goal (i.e., to retrieve denudation rate variations with a limited number of grains and a reasonable computational time), not to address very specific questions on real or synthetic cases. To this respect, this paper is extremely interesting and the Lagrangian approach used here allows to account for the large variability in individual grains histories. The effect of grid size, resolution, time step, and number of grains are tested and the results seem extremely robust. It may however reach its limits for a larger landscape (here the grid is only 10km x 10km, except for fig 7 where it is 20km x 20km), and/or with low denudation rates, for which a large number of grains would be needed.

I only have some minor to moderate comments. There are some points that are unclear to me concerning the LEM itself, independently of the CN part. Some sentences are not very clear and may need to be rephrased. Finally, I think that some outcomes deserve a deeper discussion: the influence of the grains' origin (Quartz source located close or far from the outlet) and the strong amplitude reduction of the 10Be-derived erosion rate for short period precipitation oscillations.

Thank you for these comments. We address the different points in our responses to the detailed comments.

See detailed comments in the pdf file.

Line 3:" A model could help...": awkward sentence. I suggest something like "to address this, a model can be designed to explore the statistical properties of CN concentrations in sediment grains."

Thank you. We rephrased as: "Models can be used to explore the statistics of CN concentrations in sediment grains"

Line 59: Why hillslope erosion is not simulated by a diffusion equation? Equation 3 should look like:

$$\epsilon\epsilon_h = \nabla \boxed{?} (\kappa\kappa\nabla h)$$

This is actually a pure diffusion equation if the transport length (the denominator in the deposition law) equals dx. The hillslope model in Cidre derived from the non-linear diffusion model popularized by Roering'papers, but rewritten as an erosion-deposition model. Instead of calculating the divergence between and incoming sediment flux and an outgoing flux obeying a specified transport capacity, this is the detachment flux and the deposition fluxes that are specified and the outgoing flux derives implicitly from their balance. Both formulations predict similar evolutions, what is detailed in Carretier et al. (Esurf, 2016), but the erosion-deposition model is much more stable numerically (no problem with a sediment flux going to infinity when the slope is close to the critical slope Sc) and much more adapted to the coupling with grains (Carretier et al. Esurf, 2016; EPSL, 2020).

To recall it, we added in section 2.1: "Note that the hillslope equations derive from the non-linear diffusion model (Roering et al., 1999), but written as an erosion-deposition model. Both formulations lead to the similar topographic evolution but the model used in Cidre is numerically more stable and more adapted to the coupling with grains transport (Carretier et al., 2016)".

I don't understand how drainage divides get eroded with equations 2 and 3 since basically erosion is null when the slope is null. Am I missing something?

We are not sure to understand the concern. The slope is not null from a pixel of the drainage divide to any downstream cell, so erosion can affect a cell of the divide. If the slope is null, the pixel belongs to a flat surface, for which indeed, there is no erosion.

Line 75: You use MFD to distribute the incoming water flux from the donor node to the receiver nodes, but then you only use the "steepest-descent slope" when you compute the erosion potential of the donor node.

Somehow it means that the water that is given to the other nodes does not contribute to erosion. Why not computing the erosion potential of a donor node as the sum of the contribution of each receiver node proportionally to their slope?

This is a good remark and we have been thinking a lot to that question. The way you propose was actually the algorithm in a former version of Cidre, before the mass balance was reformulated as erosion-deposition models for rivers and hillslopes. The former version was much more unstable numerically because it summed on a donor cell the non-linearities linking the erosion potentials and the water discharges in all downstream directions. The idea of calculating only one detachment potential for the donor cell according to the steepest downstream slope is that the detachment (wet or dry conditions) is mainly driven by gravity in the steepest direction. Then the water and eroded material can spread towards all the downstream directions for different reasons (approximation of shallow water equation, subcell surface rugosity etc…). The sediment routine, even the MF algorithm, is a necessary simplification in LEMs, but the exact parametrization is still a matter of research (e.g. Coatléven and Chauveau, Esurf Discussion, 2023).

Line 81: "Sediments that leave the cell are spread downstream". Do you mean "distributed to downstream cells"?

Yes. We reworded as you propose.

Line 88: Composed instead of comprised

Done.

Line 89: "They are localized by the cell number where they are located": not clear to me. Do you mean they have an index corresponding to the cell number where they were initially sown?

We rephrased as:"They are localized by the index corresponding to the cell number where they are located".

Line 98: "For a grain on a cell, it is detached if the eroded layer on that **time step** is **thicker** than or equal".

Done.

Line 110: Sediment deposition volume instead of flux; I would keep the term "flux" for something that is moving.

OK. Done.

Line 197 and 206: "Outgoing water flux" instead of "leaving water flux"

OK. Done.

Line 202:" Erode the bedrock but multiply the eroded volume by (1-sediment volume/potential erosion of sediment)" Awkward sentence. Could you rephrase it please? If the volume to be eroded is greater than the volume of sediment available, the bedrock is eroded by the remaining quantity. Is it correct?

Yes. We rephrased as: "Erode a volume of bedrock according to Equation 3 weighted by (1-sediment volume/potential erosion of sediment)"

Line 214: what do you mean by "draw the next cell"?

We rephrased as: "draw the next cell of the grain…."

Lines 215-216: Shouldn't this be in a while loop (with lines 213-214)?

We guess it could be written as a while loop equivalently.

Line 223: Do you mean user-determined output times?

We rephrased it as:"the time fits the user-defined output time".

Line 299: How do you define the "residence" time? This term is not clear to me. I would say that the "residence time" of sediments is the time spent in a given system (river network for instance) whatever they are exposed to cosmic rays or not. On the other hand, the exposure time should be the duration during

which the sediments are exposed, even partially, to cosmic rays whatever the system they reside in. Could you please be more specific on what you call the residence time?

We rephrased as: "The long residence time at shallow depth". The residence time here is the time spent by grains in the soil of the hillslopes.

Line 301: Figure 3A instead of 3B

Thank you!

Line 322: Having seen this, I think it would be interesting to see the effect of a variable Quartz source distribution (i.e., more abundant in the upstream or downstream parts of the catchment for instance) on the resulting 10Be-derived erosion rate. Maybe add this topic in the Discussion section also?

Yes! There are many other questions we want to address with this new tool, this one is a good one, but we leave this for a more thematic paper (less suitable for GMD).

Line 362: You do not discuss the amplitude. It seems that the amplitude is largely underestimated for the short period oscillations. Why is it so?

Thank for this comment. We added: "Indeed, when a grain reaches shallow depths ($<$1 m) during a low erosion rate period, its $^{10}$Be concentration is relatively high. If the grain is then rapidly exhumed, it will reach the surface with a concentration that is too high compared with what it would have been with a high rate of erosion. Once detached, if we use this concentration to determine an erosion rate, we underestimate the erosion rate. This memory effect causes the cosmogenic signal to be damped out."

---

## Author Comment (AC2)

(Responses in blue).

Carretier et al. present a timely development of their Cidre model that supports the rapidly developing field of new lab techniques for sampling cosmogenic concentrations in individual sediment grains. The model allows the numerical exploration of the landscape processes that influence the residence time of sediment grains in mountain catchments and the impact these processes have on the population statistics of cosmogenic concentrations in exported sediment. This is very useful for generating hypotheses that are becoming increasingly testable with field data.

One uncertainty that I have relates to the relationship between relief and CN production in the model and how erosion rate is defined relative to the topographic surface. It would be great if there was a figure clarifying the coordinate system for the attenuation path of cosmic rays emulated together with the surface lowering/mass removal processes.

Thank you for this suggestion. We added a Figure 1 showing the different vertical coordinates of grains as well as topographic changes during a time step in a net eroding cell and another net depositing cell. We hope that is useful. We also realized that we had used the same letter to describe cell elevation ($z$) and the grain's depth, causing confusion. We changed the grain's depth to $z'$ for its center and $z'_b$ for its basis.

Are the rays attenuated vertical or perpendicular to the topographic surface in the model? Is this important if the model is to be applicable to topographies steeper than those modelled in the paper (>30O)? I am thinking that steeper slopes do not lower their surface uniformly, so if cosmic rays are attenuated vertically, how might the production rates vary depending on the hillslope model used? E.g. non-linear hillslope diffusion model vs Cidre's model where detachment rates are proportional to slope and mass removal is modelled with non-local effects above a threshold. Some clarification would be great for us visual learners.

 The rays are attenuated vertically, namely perpendicularly to the surface of each cell and thus not perpendicular to the real surface topography approximated buy the mesh of square cells. This is not perfect; we agree and this is a concern we have. For the moment, the good match between derived and true catchment-average erosion rate shows that in average this approximation holds. For much steeper landscapes we will have to evaluate it. The DiBiase paper (2018) about the necessity to apply simple shielding correction or not depending on the topographic slope is also to be considered in the future.

Concerning the relationship with the hillslope model, there is no particular problem (steeper slope can still erode uniformly, at dynamic equilibrium in particular). The non-local hillslope model in Cidre separates the grain detachment from grain deposition, whereas the non-linear model of Roering is formulated as a difference of incoming and outgoing fluxes on a cell. Both model lead to the same evolution (cf Carretier et al., Esurf, 2016), but handling grains is far much easier in the Cidre formulation because the probabilities for a grain to be detached and then deposited are simply linked to the detachment and deposition rates on each cell. That said, the tricky part is to decide how to distribute the CN production between the grain's initial position and its final position during a time step. As the CN production rate depends on elevation and depth, this choice is important to predict the correct final CN of a grain. The strategy we explain in the manuscript ("using the mean elevation and depth of its travel during the time step.") is the best we have found to predict the correct CN concentration. The last time step before a grain becomes dead is even more critical: the CN production rate has to be "multiplied by the ratio between the depth of the grain on the starting cell of this time step and the eroded thickness on that cell during this time step" (section 3.2). If not, the final CN concentration can be wrong by more than 20% in fast eroding landscapes (because a grain can be suddenly exhumed from depth and leave the model with a two low CN concentration).

Other minor comments are included in the pdf attached.

Revisiting the grammar throughout the manuscript would do the work better justice but the paper is well structured and easy to follow. I have highlighted some of the sentences below that would benefit from clarification/rewording. The conclusion would also be made more impactful if there was a more holistic summary of the work.

Thank you very much for pointing unclear statements or proposing rephrasing. Concerning the conclusion, to keep it as short as possible, we added the sentence "The catchment-average erosion rates are approximated to within 5% uncertainty in most of the cases with a limited number of grains."

 Minor comments below:

Line2: remove 'the' from 'the relief evolution'

Done.

Line3: suggest change to 'Models can be used to explore the statistics of CN concentrations in sediment grains'

Thank you, done.

Line7: change to 'The concentrations of various CNs can be tracked in these grains.' Line10: not clear what a 'grain-by-grain distribution' is. Rephrase sentence?

Thank you, done. We cut the second part line10.

Line12: Rephrase, e.g. 'We illustrate the robustness and limitations of this approach by deriving the catchment-average erosion rates from the mean 10Be concentration of grains leaving a synthetic catchment, and comparing them to the erosion rates calculated from sediment flux, for different uplift scenarios.'

Thank you, done.

Line33: 'but without taking the evolution of the relief into account.' Could you specify why this is important?

We reworded as: "but without taking the evolution of the relief, and thus of the CN production rate into account".

Lines47, 226, 261 etc: I think the clarity of the manuscript could be improved by better defining what is meant by the 'true rate' and using this term consistently throughout the manuscript.

We added at the first occurrence: "In the following we call the 'true' average catchment erosion rate the ratio of the sediment outflux over the catchment area calculated in Cidre."

Line68: What slope threshold and transport length do you use?

0.83 and 1, respectively. We indicated these values in Table 1 to which we refer when designing the reference simulation in section 4.1

Line85: Rephrase 'they are not useful in terms of presenting the algorithm to calculate the CN concentrations in the grains'.

We rephrased it as: "because the algorithm to calculate the CN concentrations in the grains does not vary according to these processes".

Line92: Rephrase: 'For example, they can be set randomly on the grid and at depth, or with a higher density in some regions, in order to simulate the different proportions of some minerals depending on the underlying rock type.'

We rephrased it as: "For example, they can be set randomly on the grid and at depth if the grains are quartz grains and the proportion of quartz is constant in the underlying rock. Alternatively, grains can be set with a higher proportion in some cells or at some depths for which the rock has higher quartz content."

Line189: I like the pseudo code!

Thank you!

Line268: Clarify: 'In the second period, the mean erosion rate decreases to the new dynamic equilibrium value with a maximum elevation of 340 m.'

We rephrased as: "In the second period, the mean erosion rate decreases to match the lower uplift rate value at the new dynamic equilibrium. The maximum elevation is 340 m during this new equilibrium period."

Line273: different wording? 'where grains were dead...'

Reworded as: "where grains left definitively ..."

Figure 1 caption: Clarify: 'Radioactive decay slightly decreases the mean 10Be concentration calculated by Cidre, and thus the apparent inferred erosion rate neglecting radioactive decay, which is inversely proportional to the 10Be concentration, is slightly overestimated.'

We rephrased as: "The apparent inferred erosion rate is inversely proportional to the 10Be concentration (Equation 14), but because it is calculated by neglecting radioactive decay, the apparent inferred erosion rate is slightly overestimated."

Line297: interesting!

Thank you!

Line329: Why did you chose to test this variable? Include a sentence earlier in the manuscript e.g. paragraph starting line45.

We added: "As LEMs can be sensible to cell size, we tested the result of decreasing the cell size ...."

Line348: 'When the number of grains is multiplied by four, this decreases the variability (Figure 8B).' Could you expand on the significance of this? Perhaps in the discussion.

Actually, this test was just to verify a statistical fact: the more the grains, the better the average estimate (if the distribution is not heavy tailed).

Line399: rephrase 'In a Lagrangian formulation, the approach by discrete grains has advantages.'

We rephrased as: "In a Lagrangian formulation of CN concentration evolution, the approach by discrete grains has advantages"

Line433: rephrase 'and still faces the difficulty of modelling stochastic processes in a landscape evolution model'

We cut the sentence: "Nevertheless, linking a CN detrital signal with landscape evolution is not straightforward."

Line437-439: Could this be expanded on a little? I think it is an interesting part of the discussion. Could you also look at connectivity?

We are a bit afraid of expanding this part of the discussion. We agree that this is the interesting scientific part as you noticed and we are excited by the possibility to revisit the issue of paleo-denudation rates using Cidre. However, this manuscript is for a Journal describing algorithms and codes in geosciences. We are afraid that a deeper thematic discussion here may not be suitable. We left this for applications coming soon.

Line456: Reword?: 'We present a new coupling of landscape evolution model Cidre with a model of CN concentrations in individual grains.

OK thank you for your proposition.

Line458: Clarify: 'The algorithm is tested by deriving the mean catchment erosion rate from the 10Be concentration of grains leaving an uplifting catchment.' – how does this test the algorithm?

You are right. We reworded as: "The algorithm is tested by comparing the catchment-averaged erosion rate derived from the 10Be concentration of grains leaving an uplifting catchment and the true catchment-averaged erosion rate calculated by Cidre."

---

## Author Response (AR2)

Dear Dr. Carretier and co-authors,

After a careful review of the referee comments, your responses, and your revised manuscript, it is my pleasure to request a few quite small revisions. Pending these, I think that it will be ready for publication.

Thank you !

Line 64: Unindent. Also: your definition of $\epsilon_h$ should (I think) be written to note the orientation dependence of the surrounding slope. Because you are using slope (is this the absolute value of the gradient?) and you term $\kappa$ as an erodibility, it is not quite clear whether you allow deposition via this method or if (perhaps) a "pit" cell (all slopes going towards it) might still be able to lower because of the slopes around it, rather than filling in. My guess is that your code alows only erosion in this component, and checks the for downwards slopes from this cell, while allowing your deposition term to handle deposition within the cell. Perhaps you could refine this explanation?

OK we added :
« S is the slope (absolute value of the elevation gradient) towards the downstream cell in the steepest direction, (...).  There is no erosion in a pit cell, only deposition.«

Line 340: sensible --> sensitive. (I always have to check myself in Latin languages, from the other side of this!)
Thank you.

Figure 3: It looks like the "dynamic equilibrium" labels are applied to times at which the landscape is still approaching, but close to, a dynamic equilibrium. This comment from me is because I am teaching Geomorphology right now, and based on student questions, am feeling a push towards precision in our language.
OK we reworded as « close to dynamic equilibrium. »

Line 287: left the model grid definitively
Added.

I am very glad for Figure 7 (scale impacts) and considerations of transience.
Thank you !

We have also checked that colors in our figures  allow readers with colour vision deficiencies to correctly interpret our finding.

Best wishes,
Andy Wickert

Additional private note (visible to authors and reviewers only):
Dear Dr. Carretier and co-authors,

In addition to the above note, I wanted to share my deep apology for the time that this response has taken. I was managing a back injury and a COVID infection between the time of your resubmission and now. I am going as quickly as I can through the backlog.

No problem, time is an issue for all of us. Thank you for this editing work.  We wish you a rapid recovery.

Thank you for your diligent work to produce a high-quality manuscript; I am glad to see the explicit inclusion of measurable quantities in our landscape-evolution models.

Best wishes,
Andy